# Readmission and emergency department presentation after hospitalisation for epilepsy in people with intellectual disability: A data linkage study

Peiwen Liao[1], Claire M. Vajdic[2¤*], Simone Reppermund[1,3], Rachael C. Cvejic[1], Tim R. Watkins[1], Preeyaporn Srasuebkul[1], Julian Trollor[1,3]

1 Faculty of Medicine and Health, Department of Developmental Disability Neuropsychiatry, Discipline of Psychiatry and Mental Health, University of New South Wales, Sydney, Australia, 2 Faculty of Medicine and Health, Centre for Big Data Research in Health, University of New South Wales, Sydney, Australia, 3 Faculty of Medicine and Health, Centre for Healthy Brain Ageing, Discipline of Psychiatry and Mental Health, University of New South Wales, Sydney, Australia

¤ Current address: Faculty of Medicine and Health, Kirby Institute, University of New South Wales, Sydney, Australia

* claire.vajdic@unsw.edu.au

**Data Availability Statement:** The original data are not publicly accessible due to the conditions of use imposed by the data custodians and ethics

## Abstract

### Background

Despite the high prevalence of epilepsy and multiple barriers to care in people with intellectual disability, the risk of returning to hospital after an admission for epilepsy is largely unknown. In this study, we sought to quantify and compare readmission and emergency department (ED) presentations after hospitalisation for epilepsy in people with and without intellectual disability.

### Methods and findings

Using linked administrative datasets, we conducted a retrospective cohort study of people aged 5–64 years with an acute hospitalisation for epilepsy from 2005–2014 in New South Wales, Australia. Acute readmission and ED presentation rates within 30, 90, and 365 days of the index hospitalisation were estimated and compared between people with and without intellectual disability using modified Poisson regression. Of 13537 individuals with an index hospitalisation, 712 children and 1862 adults had intellectual disability. Readmission and ED presentation after the index hospitalisation were common in people with intellectual disability. Within 30 days, 11% of children and 15.6% of adults had an all-cause readmission and 18% of children and 23.5% of adults had an ED presentation. Over 60% of both children and adults presented to an ED within a year. Neurological, respiratory, and infectious conditions were overrepresented reasons for readmission in people with intellectual disability. Age-adjusted relative risks (RRs) within each period showed a higher risk of readmission and ED presentation in children and adults with intellectual disability than without. Most RRs remained statistically significant after controlling for covariates. The largest adjusted RRs

committee. Access to the data and analytical files is permitted subject to the approval of the human research ethics committees and data custodians. Researchers interested in applying for data access or collaboration should contact the Department of Developmental Disability Neuropsychiatry (dddn@unsw.edu.au) with their expression of interest.

**Funding:** This work was supported by National Health and Medical Research Council grant (NHMRC) (Grant name: Partnership Project APP1056128 and Project Grant APP1123033 to CV, JT, and SR; URLs: https://www.nhmrc.gov.au/funding); and UNSW Scientia PhD Scholarship (to PL; URLs: https://www.scientia.unsw.edu.au/). The funders had no role in study design, data collection and analysis, decision to publish, or preparation of the manuscript

**Competing interests:** The authors have declared that no competing interests exist.

were observed for readmission for epilepsy (RR 1.70, 95% CI: 1.42 to 2.04) and non-epilepsy related conditions (RR 1.73, 95%: CI 1.43 to 2.10) in children. Study limitations include lack of clinical data.

## Conclusions

Increased risk of returning to acute care after epilepsy hospitalisation suggests there is a need to improve epilepsy care for people with intellectual disability. We recommend research into strategies to improve management of both seizures and comorbidity.

## Introduction

Intellectual disability, defined by impairments in cognitive and adaptive functioning with onset during the developmental period [1, 2], affects approximately 1–2% of the population [3]. People with intellectual disability experience high rates of several health conditions including epilepsy [4]. A recent meta-analysis [5] found that 22% of people with intellectual disability have epilepsy, markedly higher than in the general population (<1.0%) [6].

Challenges in epilepsy management for people with intellectual disability include communication difficulties, high risk of refractory and frequent seizures, and neuropsychiatric comorbidities [7–10]. People with intellectual disability and epilepsy are at greater risk of hospitalisation and emergency department (ED) presentation than people with epilepsy or intellectual disability alone [11–14], and hospitalisation for epilepsy accounts for a large proportion of potentially avoidable hospitalisation for people with intellectual disability [15]. Individuals with both epilepsy and intellectual disability also experience other poor health outcomes, including increased risk of mortality [16].

Following hospitalisation for epilepsy, an important outcome is readmission. Repeated hospitalisations can induce psychological distress [17] and are costly [18]. They are thought to reflect suboptimal or poor continuity of care [19–21]. In the general population, the average unplanned (or acute) readmission rate within 30 days of epilepsy- or seizure-related hospitalisations is 10%, with epilepsy or seizure being the most common indication [20, 21]. Other common reasons for readmission within a year include psychiatric disorders [22, 23] and suicide attempts [24]. Few studies have reported readmission after epilepsy hospitalisation in people with intellectual disability, with one observing an increased risk of readmission in children with intellectual disability compared to children without [20]. However, the aforementioned study relied solely on hospital records to identify intellectual disability status, probably resulting in a sample with more severe disability or complex health needs [25]. Further research is needed to help understand how intellectual disability affects readmission risk after epilepsy hospitalisation.

This study aimed to compare rates of unplanned readmissions and ED presentations after hospitalisation for epilepsy among children and adults with and without intellectual disability. We also examined readmission for epilepsy and non-epilepsy conditions, specifically psychiatric disorders.

## Methods

### Data sources

We extracted data from an established linked dataset containing health and services records of people with neuropsychiatric disorders [25]. The data were drawn from multiple population-

based administrative datasets from New South Wales (NSW), Australia as previously described [25]. Key datasets for this study were the Admitted Patient Data Collection (APDC), Emergency Department Data Collection (EDDC) and Registry of Births, Deaths and Marriages death records.

## Study population

Our study population was selected from the APDC, which contains all public and private hospital admissions in NSW. Principal and additional diagnoses are recorded for each admission using the International Statistical Classification of Diseases and Related Health Problems, Tenth Revision, Australian Modification (ICD-10-AM) [2].

Fig 1 describes how the cohort was derived. From all people who were discharged from hospital during the study period, we included people discharged from an unplanned acute care admission between 1 July 2005 and 30 June 2014 where the principal diagnosis recorded was epilepsy (G40) or status epilepticus (G41). S1 Table documents how we defined 'unplanned' and 'acute'. The first admission was defined as the index admission. After excluding individuals with implausible death or age records, we excluded individuals if on discharge from the index admission they i) died, were transferred to hospice care, left hospital against advice, or had no discharge mode recorded [21]; ii) were younger than five or older than 64 years; or iii) had missing covariate data. The age limits were chosen to avoid bias introduced by young children yet to receive an intellectual disability diagnosis and older people for whom our data capture may be incomplete due to transition to aged care services [26]. Excluding older adults also reduced the likelihood of inclusion of epilepsy associated with late life disorders like dementia [27], which may complicate service use.

Diagnoses of intellectual disability were drawn from multiple health and disability datasets as previously described [25], by which people classified as having intellectual disability must have a diagnosis based on the Diagnostic and Statistical Manual of Mental Disorders IV or ICD-10 [1, 2]. This strategy resulted in formation of a cohort of people with intellectual disability who needed medical or disability service supports in NSW, representing about 1% of the NSW population in 2015 [25], comparable to the average estimate worldwide [3]. The remainder of the cohort was classified as not having intellectual disability.

We matched people with and without intellectual disability on age and sex using Coarsened Exact Matching to balance the distribution of covariates [28]. We formed strata defined by age group (2-years for children <16 years and 5-years for adults) and sex, with a weight applied to each individual's observations to account for the number of people in their strata. This matching procedure retained all individuals in the analysis.

## Outcomes

We identified outcomes from the hospitalisation (APDC) and ED (EDDC) datasets. The EDDC contains information on ED presentations in 60% of NSW public hospitals between 1 July 2005 and 30 June 2016 [29]. The primary outcomes were all-cause unplanned hospital readmissions and ED presentations from the day of discharge from the index admission until day 30, day 90, day 365, or until death if that occurred first. Secondary outcomes were unplanned readmissions for epilepsy (G40-G41) and non-epilepsy conditions (G40 and G41 excluded), and specifically psychiatric disorders (F00-F69 and F80-F99), based on the principal diagnosis at discharge.

We did not include readmissions on the day of discharge from the index admission.

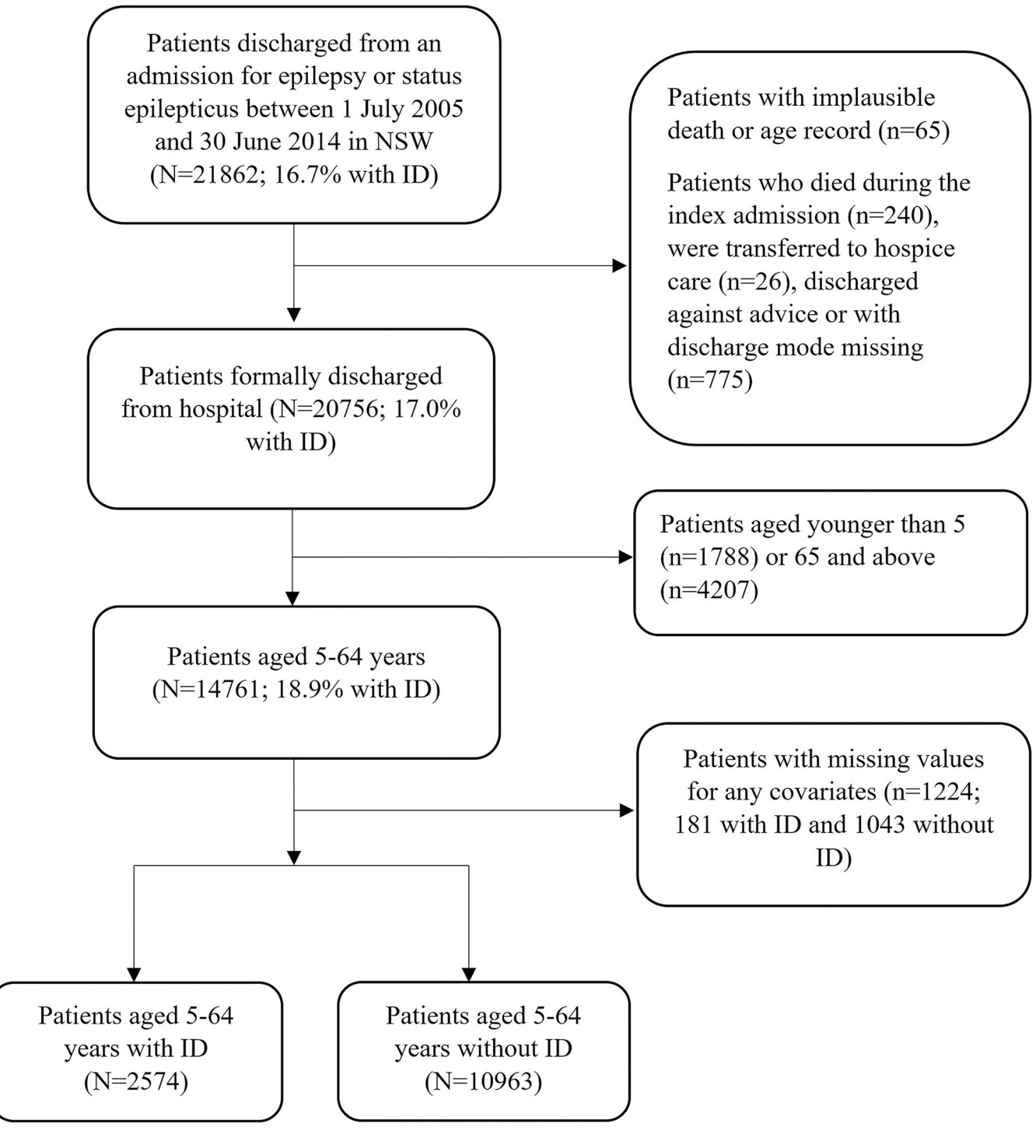

ID: intellectual disability

**Fig 1. Cohort derivation flowchart.**

## Statistical analyses

All analyses were conducted separately for children (5–15 years) and adults (16–64 years) given the expected differences in the populations (e.g., some types of epilepsy), outcomes, and the age of transition from paediatric to adult inpatient services in NSW, Australia.

Descriptive statistics were used to compare demographic characteristics, index admission characteristics, and comorbidities for people with and without intellectual disability. We calculated the proportions of people with one or more readmissions or ED presentations during each follow-up period. The principal diagnosis of the first all-cause readmission was grouped by ICD-10-AM disease chapter. We also calculated the median number of days to the first outcome. T-test, Chi-square test (or Fisher exact test if ≤ five persons in one group), or Mann Whitney U test was used as applicable.

We fitted modified Poisson regression with robust estimation to examine associations between intellectual disability and the outcomes [30]; a survival analysis could not be used because the proportional hazards assumption was violated. We first fitted a model only adjusting for age at discharge from the index admission (not applicable for children) and then a model with select covariates derived from the APDC dataset. Age, sex, socio-economic status, index admission characteristics, and comorbidities are associated with intellectual disability or acute hospital visits after epilepsy hospitalisation [3, 21, 31]. Therefore, we selected age, sex, country of birth (Australia, overseas), Index of Relative Socio-economic Disadvantage (IRSD; based on area of residence), remoteness of residential area (Accessibility and Remoteness Index of Australia), hospitalisations in the previous year, characteristics of the index admission (length of stay, hospital type [public, private], mode of discharge [community, other accommodation or nursing home]), and comorbidity status as covariates. For comorbidity status, we retrieved all diagnoses recorded at the index admission and admissions in the previous year to obtain the Charlson comorbidity index (CCI) [32] and psychiatric diagnoses. The latter excluded intellectual disability and dementia. The CCI and psychiatric disorders were not included as covariates in the modelling for children due to the low prevalence of these conditions in this age group. The most common neuropsychiatric disorders in children with intellectual disability are other neurodevelopmental conditions (e.g., autism and attention deficit hyperactivity disorder) [33], which are less common in the general population [34] and, more importantly, may lie on the causal pathway.

In the main analysis, we excluded individuals with missing data (less than 10%), as the cohort characteristics before and after the exclusion were comparable. We also performed a sensitivity analysis to test the impact of excluding people with missing data. The level of significance was set to $p < 0.05$. We performed data matching and analyses in Stata 15.

## Ethics approval

This study was approved by NSW Population & Health Services Research Ethics Committee, and access to the data sets was granted by relevant data custodians. As the research met specific safeguards, the requirement for informed consent was waived by the presiding ethics committee.

## Results

### Cohort characteristics

Out of 21862 patients with an admission for epilepsy during the study period, 20756 patients were formally discharged from the hospital, among which, we included all patients aged 5–64 years without missing records for any covariates (2574 with intellectual disability; Fig 1). This

final cohort consisted of 2104 children and 11433 adults. Children had a higher prevalence of intellectual disability than adults (33.8% versus 16.3%; Table 1). Cohort characteristics and outcomes before and after excluding participants with missing data showed that missingness was unrelated to the outcomes (characteristics of participants with complete data shown in S2 Table).

**Table 1. Cohort characteristics at the index admission for children and adults with and without Intellectual disability (ID) (n, %).**

| | Children (N = 2,104) | | | Adults (N = 11,433) | | |
|---|---|---|---|---|---|---|
| | ID (N = 712) | Non-ID (N = 1,392) | P value[c] | ID (N = 1,862) | Non-ID (N = 9,571) | P value[c] |
| **Male** | 409 (57.4) | 743 (53.4) | 0.076 | 1064 (57.1) | 5488 (57.3) | 0.875 |
| **Median (IQR) age (years)** | 9.5 (7–12) | 10.8 (8–11) | | 36.7 (25–49) | 39.8 (28–51) | |
| **Age range (years)** | | | | | | |
| 16–24 | N/A | N/A | | 469 (25.2) | 1754 (18.3) | <0.001 |
| 25–44 | N/A | N/A | | 786 (42.2) | 4166 (43.5) | |
| 45–64 | N/A | N/A | | 607 (32.6) | 3651 (38.2) | |
| **Born in Australia** | 674 (94.7) | 1289 (92.6) | 0.073 | 1735 (93.2) | 7743 (80.9) | <0.001 |
| **Remoteness of residence** | | | | | | |
| Major city | 469 (65.9) | 939 (67.5) | 0.348 | 1225 (65.8) | 6419 (67.1) | 0.389 |
| Inner regional | 165 (23.2) | 328 (23.6) | | 442 (23.7) | 2240 (23.4) | |
| Outer regional/remote/very remote | 78 (11.0) | 125 (9.0) | | 195 (10.5) | 912 (9.5) | |
| **Index of relative socio-economic disadvantage based on place of residence** | | | | | | |
| 1–2 (most disadvantaged) | 143 (20.1) | 366 (26.3) | <0.001 | 366 (19.7) | 2390 (25.0) | <0.001 |
| 3–4 | 148 (20.8) | 280 (20.1) | | 284 (20.6) | 2107 (22.0) | |
| 5–6 | 161 (22.6) | 278 (20.0) | | 409 (22.0) | 1865 (19.5) | |
| 7–8 | 150 (21.1) | 204 (14.7) | | 417 (22.4) | 1590 (16.6) | |
| 9–10 (least disadvantaged) | 110 (15.5) | 264 (19.0) | | 286 (15.4) | 1619 (16.9) | |
| **Characteristics of the index admission** | | | | | | |
| **Private health insurance** | 179 (25.1) | 419 (30.1) | 0.017 | 205 (11.0) | 2020 (21.1) | <0.001 |
| **Public hospital** | >700 (>95)[a] | 1374 (98.7) | 0.024 | 1853 (99.5) | 9485 (99.1) | 0.071 |
| **Median (IQR) length of stay (days)** | 1.0 (1–2) | 1.0 (1–2) | | 1.0 (1–3) | 1.0 (1–3) | |
| **Length of stay (days)** | | | | | | |
| 1–2 | 542 (76.1) | 1186 (85.2) | <0.001 | 1284 (69.0) | 7098 (74.2) | <0.001 |
| 3–6 | 120 (16.9) | 176 (12.6) | | 383 (20.6) | 1617 (16.9) | |
| ≥7 | 50 (7.0) | 30 (2.2) | | 195 (10.5) | 856 (8.9) | |
| **Mode of separation** | | | | | | |
| Discharge by hospital | >700 (>95)[a] | >1350 (>95)[a] | 1.000 | 1828 (98.2) | 9496 (99.2) | <0.001 |
| Transfer to nursing home | N/A | N/A | | 18 (1.0) | 58 (0.6) | |
| Transfer to other accommodation | <5[a] | <5[a] | | 16 (0.9) | 17 (0.2) | |
| **≥ one admission in the year prior** | 369 (51.8) | 399 (28.7) | <0.001 | 897 (48.2) | 4215 (44.0) | 0.001 |
| **Charlson Comorbidity Index** | | | | | | |
| 0 | 671 (94.2) | 1338 (96.1) | 0.126 | 1613 (86.6) | 7990 (83.5) | <0.001 |
| 1–2 | 35 (4.9) | 48 (3.5) | | 190 (10.2) | 950 (9.9) | |
| ≥3 | 6 (0.8) | 6 (0.4) | | 59 (3.2) | 631 (6.6) | |
| Psychiatric comorbidity[b] | 269 (37.8) | 82 (5.9) | <0.001 | 536 (28.8) | 2697 (28.2) | 0.590 |

[a] True value and/or percentage were censored to ensure confidentiality.

[b] ICD-10 codes for psychiatric comorbidity: F00-F99, except for intellectual disability (F70-F79) and dementia (F00-F03 or F05.1; according to the codes included in the Charlson Comorbidity Index).

[c] P value was estimated based on independent t-test for continuous variables and Chi-squared test for categorical variables.

Compared to adults without intellectual disability, a higher proportion of adults with intellectual disability were born in Australia (P<0.001). Individuals with intellectual disability, regardless of age, were more likely to live in moderately socioeconomically disadvantaged areas (P<0.001).

At the index admission, almost all people attended a public hospital and were discharged to the community. Therefore, we did not include the hospital and discharge type variable in the regression model in the next stage. Compared to people without intellectual disability, people with intellectual disability were less likely to hold private health insurance and more likely to have longer hospital stays (≥3 days) and an admission in the previous year (P≤0.001). Compared to adults without intellectual disability, adults with intellectual disability were slightly more likely to have a CCI of 0. Psychiatric comorbidities were common in all adults (28.8% and 28.2% for those with and without intellectual disability, respectively) and children with intellectual disability (37.8%), but not in children without (5.9%). As noted in the statistical analyses section, the difference in prevalence of psychiatric comorbidities in children with and without intellectual disability may primarily be due to the higher risk of neurodevelopmental disorders in children with intellectual disability. Neurodevelopmental conditions (e.g. autism and attention deficit hyperactivity disorder) are typically diagnosed early in life, whereas other mental health conditions (e.g. depression and anxiety) are more likely to be diagnosed during young adulthood [35].

## All-cause readmissions

Compared to people without intellectual disability, a larger proportion of people with intellectual disability were readmitted to hospital (Table 2). Seventy-eight (11.0%) children and 291 (15.6%) adults with intellectual disability had at least one readmission within 30 days. The proportions increased to 46.4% (children) and 52.8% (adults) within a year. Almost one third of people with intellectual disability had more than one all-cause readmission within a year, compared to 9.4% of children and 20.7% of adult comparators. There was no difference in the number of days to the first readmission between people with and without intellectual disability (Table 3).

The dominant reason for the first readmission in people with and without intellectual disability was neurological disorders (>95% of which were epilepsy), followed by unspecified symptoms (Fig 2). Neurological conditions were slightly over-represented in adults with intellectual disability within each follow-up period. There was a larger proportion of readmissions for respiratory conditions in adults and children with intellectual disability (particularly lung diseases due to external agents, influenza, and pneumonia; the latter two were only overrepresented in children), and infectious diseases in children with intellectual disability, than those without.

The age-adjusted model showed a higher risk of all-cause readmission in people with intellectual disability compared to people without intellectual disability within each follow-up period. After adjusting for other covariates, the increased risk remained significant for all except 30-day readmission in children. The highest relative risk was found for children within 365 days (RR 1.58, 95%CI: 1.39 to 1.79).

## Readmissions for specific reasons

Compared to people without intellectual disability, higher proportions of people with intellectual disability were readmitted for both epilepsy and non-epilepsy conditions across the follow-up periods, but the median times to readmission were comparable (Tables 2 and 3). Psychiatric readmission rates were generally low and similar between groups.

**Table 2. Comparison of the proportions of people with and without intellectual disability with at least one outcome within each follow-up period by age.**

| Outcomes | Children | | | | Adults | | | |
|---|---|---|---|---|---|---|---|---|
| | n (%) | | RR (95% CI) | | n (%) | | RR (95% CI) | |
| | ID | Non-ID | Model 1[a] | Model 2[b] | ID | Non-ID | Model 1[a] | Model 2[b] |
| **All-cause readmission** | | | | | | | | |
| Within 30 days | 78 (11.0) | 98 (7.0) | 1.65 (1.23–2.21) | 1.26 (0.94–1.69) | 291 (15.6) | 1011 (10.6) | 1.51 (1.34–1.70) | 1.42 (1.26–1.61) |
| Within 90 days | 157 (22.1) | 191 (13.7) | 1.75 (1.44–2.14) | 1.42 (1.15–1.74) | 506 (27.2) | 1974 (20.6) | 1.35 (1.24–1.47) | 1.27 (1.17–1.39) |
| Within 365 days | 328 (46.1) | 376 (27.0) | 1.81 (1.60–2.05) | 1.58 (1.39–1.79) | 983 (52.8) | 3907 (40.8) | 1.33 (1.26–1.40) | 1.26 (1.19–1.32) |
| **Readmission for epilepsy** | | | | | | | | |
| Within 30 days | 49 (6.9) | 64 (4.6) | 1.57 (1.09–2.28) | 1.25 (0.86–1.81) | 143 (7.7) | 419 (4.4) | 1.71 (1.42–2.06) | 1.64 (1.36–1.97) |
| Within 90 days | 90 (12.6) | 118 (8.5) | 1.63 (1.25–2.13) | 1.43 (1.08–1.88) | 267 (14.3) | 869 (9.1) | 1.55 (1.36–1.76) | 1.47 (1.29–1.68) |
| Within 365 days | 209 (29.4) | 230 (16.5) | 1.90 (1.60–2.25) | 1.70 (1.42–2.04) | 572 (30.7) | 1907 (19.9) | 1.53 (1.41–1.66) | 1.44 (1.33–1.56) |
| **Readmission for non-epilepsy conditions** | | | | | | | | |
| Within 30 days | 34 (4.8) | 38 (2.7) | 1.98 (1.24–3.18) | 1.36 (0.84–2.18) | 170 (9.1) | 643 (6.7) | 1.42 (1.21–1.68) | 1.33 (1.13–1.57) |
| Within 90 days | 90 (12.6) | 89 (6.4) | 2.18 (1.63–2.92) | 1.54 (1.14–2.08) | 305 (16.4) | 1302 (13.6) | 1.27 (1.14–1.43) | 1.20 (1.07–1.35) |
| Within 365 days | 200 (28.1) | 192 (13.8) | 2.21 (1.84–2.66) | 1.73 (1.43–2.10) | 651 (35.0) | 2827 (29.5) | 1.24 (1.16–1.33) | 1.16 (1.09–1.25) |
| **Readmission for psychiatric disorders** | | | | | | | | |
| Within 30 days | <5[c] | 5 (0.4) | N/A[d] | N/A[d] | 40 (2.2) | 140 (1.5) | 1.52 (1.07–2.16) | 1.47 (1.03–2.10) |
| Within 90 days | 7 (1.0) | 9 (0.7) | N/A[d] | N/A[d] | 76 (4.1) | 291 (3.0) | 1.41 (1.10–1.80) | 1.35 (1.05–1.73) |
| Within 365 days | 12 (1.7) | 22 (1.6) | N/A[d] | N/A[d] | 161 (8.7) | 695 (7.3) | 1.23 (1.04–1.44) | 1.14 (0.97–1.33) |
| **ED presentation** | | | | | | | | |
| Within 30 days | 128 (18.0) | 214 (15.4) | 1.23 (1.00–1.51) | 1.03 (0.84–1.27) | 438 (23.5) | 1628 (17.0) | 1.37 (1.25–1.50) | 1.29 (1.18–1.42) |
| Within 90 days | 241 (33.9) | 394 (28.3) | 1.26 (1.10–1.45) | 1.08 (0.94–1.25) | 758 (40.7) | 3012 (31.5) | 1.29 (1.21–1.37) | 1.21 (1.14–1.29) |
| Within 365 days | 456 (64.0) | 697 (50.1) | 1.31 (1.22–1.42) | 1.23 (1.13–1.33) | 1264 (67.9) | 5477 (57.2) | 1.19 (1.15–1.23) | 1.14 (1.10–1.18) |

[a] Model 1 was adjusted for age for adults and unadjusted for children.

[b] Model 2 was adjusted for age, sex, country of birth, residential remoteness, IRSD, private health insurance at index admission, length of stay at index admission, Charlson comorbidity index, comorbid psychiatric disorders and admission during the last year for adults; age, Charlson Comorbidity Index, and psychiatric comorbidities were excluded from the model for children.

[c] True value and percentage were censored to ensure confidentiality.

[d] N/A: Readmissions for psychiatric disorders are not applicable to children.

The age-adjusted model showed a higher risk of epilepsy and non-epilepsy readmission in people with intellectual disability than those without in each follow-up period. In the multivariable model, the increased risk remained significant for all except 30-day epilepsy or non-epilepsy readmission in children. The highest risk was observed for epilepsy and non-epilepsy readmission in children within a year (RR 1.70, 95% CI: 1.42 to 2.04; RR 1.73, 95% CI: 1.43 to 2.10, respectively).

The age-adjusted model also showed an increased risk of psychiatric readmission within each follow-up period in adults with intellectual disability compared to those without (Table 2), which remained significant for readmissions within 30 and 90 days after adjusting for covariates. The largest RR was observed within 30 days of the index admission (RR 1.47, 95% CI: 1.03 to 2.10).

## ED presentations

In both observation periods (30 and 365 days), a greater proportion of people with intellectual disability presented to ED compared to those without (Table 2). Almost half the children and adults with intellectual disability had more than one ED presentation within 365 days, versus around one-third of the comparison groups. Within 365 days, the median number of days to

**Table 3. Days to the first outcome (median, IQR) within each follow-up period for people with and without intellectual disability with at least one outcome.**

| Outcomes | Children | | | | | Adults | | | | |
|---|---|---|---|---|---|---|---|---|---|---|
| | ID | | Non-ID | | | ID | | Non-ID | | |
| | n[a] | Median (IQR) | n[a] | Median (IQR) | P value[c] | n[a] | Median (IQR) | n[a] | Median (IQR) | P value[c] |
| **All-cause readmissions** | | | | | | | | | | |
| Within 30 days | 78 | 9 (4–21) | 98 | 9 (4–18) | 0.695 | 291 | 9 (4–19) | 1011 | 11 (4–20) | 0.037 |
| Within 90 days | 157 | 31 (9–59) | 191 | 29 (8–53) | 0.622 | 506 | 25 (7–51) | 1974 | 30 (11–55) | 0.003 |
| Within 365 days | 328 | 97 (33–203) | 376 | 90 (28–205) | 0.363 | 983 | 83 (24–194) | 3907 | 88 (29–193) | 0.149 |
| **Readmission for epilepsy** | | | | | | | | | | |
| Within 30 days | 49 | 12 (4–23) | 64 | 12 (4–23) | 0.451 | 143 | 9 (3–30) | 419 | 11 (4–20) | 0.778 |
| Within 90 days | 90 | 27 (12–51) | 118 | 27 (12–51) | 0.844 | 267 | 28 (8–55) | 869 | 32 (11–58) | 0.128 |
| Within 365 days | 209 | 113 (34–223) | 230 | 113 (34–223) | 0.056 | 572 | 101 (31–210) | 1907 | 106 (37–204) | 0.517 |
| **Readmission for non-epilepsy conditions** | | | | | | | | | | |
| Within 30 days | 34 | 8 (5–17) | 38 | 9.5 (4–15) | 0.964 | 170 | 10 (4–18) | 643 | 11 (5–21) | 0.077 |
| Within 90 days | 90 | 43.5 (14–68) | 89 | 37 (11–56) | 0.214 | 305 | 26 (9–51) | 1302 | 31 (12–57) | 0.011 |
| Within 365 days | 200 | 106 (49–220.5) | 192 | 104.5 (38–217) | 0.466 | 651 | 103 (28–212) | 2827 | 106 (35–211) | 0.228 |
| **Readmission for psychiatric disorders** | | | | | | | | | | |
| Within 30 days | <5[b] | 15 (4–17) | 5 | 15 (7–18) | 0.764 | 40 | 11 (5–21) | 140 | 13 (7–23) | 0.336 |
| Within 90 days | 7 | 56 (15–77) | 9 | 22 (15–53) | 0.340 | 76 | 29 (10–49) | 291 | 32 (14–57) | 0.163 |
| Within 365 days | 12 | 79 (37–225) | 22 | 114 (52–237) | 0.829 | 161 | 94 (31–212) | 695 | 113 (40–227) | 0.160 |
| **ED presentation** | | | | | | | | | | |
| Within 30 days | 128 | 8 (3–18) | 214 | 8 (4–17) | 0.965 | 438 | 8 (3–16) | 1628 | 9 (3–18) | 0.268 |
| Within 90 days | 241 | 27 (8–54) | 394 | 25 (8–53) | 0.765 | 758 | 24 (7–48) | 3012 | 27 (8–53) | 0.018 |
| Within 365 days | 456 | 81 (26–182) | 697 | 73 (21–175) | 0.208 | 1264 | 61 (15–145) | 5477 | 75 (23–177) | <0.001 |

[a] The number of patients having at least one outcome (readmission or ED presentation).

[b] True value was censored to ensure confidentiality.

[c] P value was estimated based on Mann Whitney U.

the first ED presentation was shorter for adults with intellectual disability than those without (Table 3).

The age-adjusted RR showed a significantly increased risk of ED presentations in adults with intellectual disability within all follow-up periods and in children with intellectual disability within 90 and 365 days (Table 2). After adjusting for covariates, all RRs remained significant for adults and only the RR within 365 days remained significant for children. The largest RR was observed for adults within 30 days (RR 1.29, 95% CI: 1.18 to 1.42).

## Discussion

In this population-based study using linked administrative datasets, we showed re-entry to acute care after epilepsy hospitalisation was common, especially in adults with intellectual disability. Intellectual disability status was associated with a higher readmission and ED presentation risk within 30, 90, and 365 days of the index hospitalisation. The risk of readmission for both epilepsy and non-epilepsy conditions was elevated. Over-represented causes of readmission included neurological and respiratory conditions in all people with intellectual disability, and infectious disorders in children with intellectual disability. These findings suggest there is a need to provide specialised and interdisciplinary support for inpatient and post-acute care support around epilepsy management for people with intellectual disability.

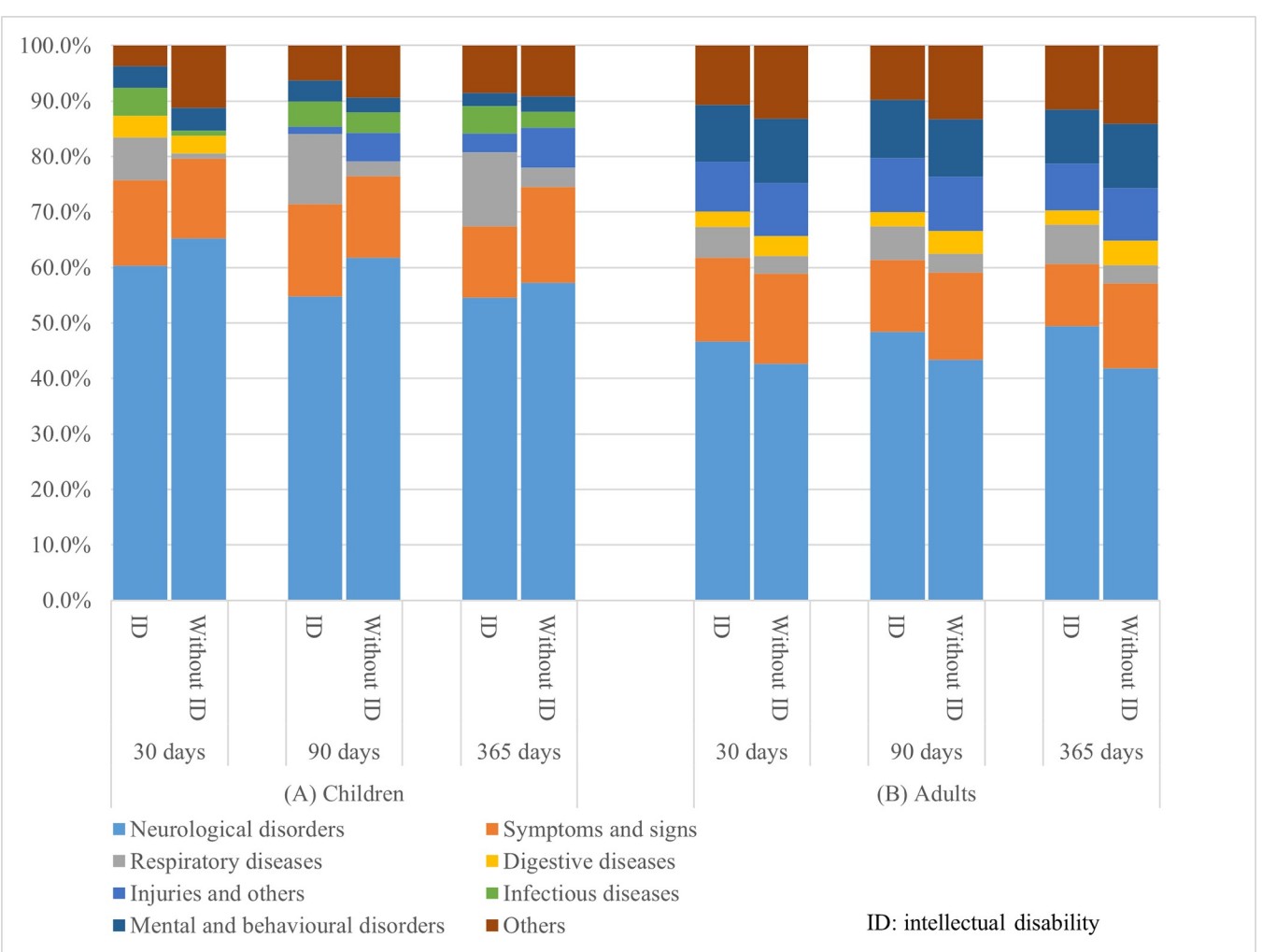

**Fig 2.** The six most frequent primary diagnoses at the first readmission within each follow-up period for children (A) and adults (B) by intellectual disability status.

Our study confirmed prior evidence showing a high rate of 30-day all-cause readmission in children with intellectual disability following epilepsy hospitalisation [20]. Expanding the previous knowledge, we further showed high readmission and ED presentation rates within 30, 90, and 365 days in both children and adults with intellectual disability. The proportion of people readmitted was higher in adults than children with intellectual disability, resembling the trend in the general population [31]. We also found a high rate of repeated ED presentations within a year in people with intellectual disability, possibly indicating inadequate acute seizure management. Our findings highlight the need to address the determinants of frequent ED use in people with epilepsy and intellectual disability. Collectively, these results show a high rate of acute care utilisation after discharge from epilepsy hospitalisation among people with intellectual disability.

Intellectual disability was associated with increased risk of readmission, for both epilepsy and non-epilepsy conditions, and ED presentation within the short- and long-term in adults and within the long-term in children. We observed no association between 30-day all-cause readmission and intellectual disability status in children, contrasting with the 50% excess

readmission risk reported in an American cohort (1–17 years) using a hospital register [20]. The discordant finding may reflect differences in health systems between two countries, or may have occurred due to methodological differences. We included only children aged 5–15 years, adjusted for previous hospitalisations as a potential indicator of general health status, and identified intellectual disability from multiple reliable sources.

The association between higher risk of acute care and intellectual disability was independent of sociodemographic, health, and index admission characteristics in this study. We surmise that factors that have not been controlled for, including epilepsy characteristics and potentially inappropriate or inadequate management, may have driven the excess acute care in people with intellectual disability. More severe and frequent seizures often occurring with intellectual disability could result in more inpatient care, which may be entirely appropriate. However, previous research has revealed multiple gaps in epilepsy care for people with intellectual disability, including a low level of multidisciplinary care [36] and inadequate use of evidence-based treatments [37]. Bridging these gaps by introducing specialised services with expertise in intellectual disability and epilepsy, while mitigating the general barriers people with intellectual disability experience in seeking quality health care [38, 39], has the potential to reduce seizure frequency and improve the comorbidity management, thus reducing the need for recurrent emergency hospital care.

We observed variations in relative risks of readmission over time since the index hospitalisation between children and adults. Generally, the disparity in readmission rates between those with and without intellectual disability was the largest within 30 days and decreased over time in adults, while it was the lowest within 30 days and peaked within one year in children. As 30-day all-cause readmission is widely used to indicate the quality of inpatient care, and one-year readmission might be partly attributed to outpatient care after discharge, this finding may suggest care gaps are more related to inpatient care for adults and post-discharge care for children.

Epilepsy or seizure, the leading reason for readmission in the general population with epilepsy [20, 21], accounted for an even larger proportion of readmissions in people with intellectual disability. This suggests that there is a need for improved clinical epilepsy care pathways in this high-risk patient group, which might emphasise continuity of care in the community, more frequent reviews of management, and access to highly specialised reviews for those with complex epilepsy and health needs. Disability sector staff could also be better equipped to support seizure management, including timely access to appropriate health experts. We also observed an overrepresentation of readmissions for non-epilepsy conditions in people with intellectual disability. Hospitalisation for respiratory disorders, specifically influenza, pneumonia, and lung diseases due to external agents, is consistent with evidence of more readmissions for respiratory diseases in people with intellectual disability [40, 41]. This suggests a need for better influenza and aspiration prevention in their epilepsy care. The excess readmission for infectious diseases within 30 days in children was unexpected, which may reflect more undetected underlying central nervous system infection [42]. We did not observe more readmissions for injuries in people with intellectual disability despite evidence of an association between injuries and intellectual disability from a Scottish cohort [43], suggesting injuries in our cohort were mainly related to epilepsy rather than disability.

Consistent with prior evidence of an association between neuropsychiatric disorders and intellectual disability in people with epilepsy [10], we found an association between intellectual disability and psychiatric readmission risk within 30 and 90 days in adults. The association was most pronounced within 30 days, suggesting the need to improve screening and enhance monitoring for neuropsychiatric symptoms during and shortly after epilepsy hospitalisations, respectively, in adults with intellectual disability.

We used population-based databases to build one of the largest intellectual disability cohorts internationally. Combining health and disability registers enabled a more representative cohort than relying on one register alone [20], thus yielding more generalisable results. The data linkage allowed us to ascertain outcomes systematically and consistently for all participants and to account for death. Study limitations include potential coding and recording errors and lack of clinical information in the administrative datasets, particularly the severity of epilepsy and intellectual disability. While we were not able to ascertain disability severity, we have generated evidence on the needs of people with intellectual disability as a whole and identified areas that could benefit from further research. Furthermore, as the diagnosis of epilepsy not otherwise specified was common, we were unable to explore variation by epilepsy subtype. Additionally, although the residential setting could influence ED visits in people with intellectual disability [44], we did not have complete data on the residential setting or living arrangements. Finally, while the participating EDs are large centres that serve a substantial proportion of the NSW population, the lack of population coverage of ED presentations means that we have underestimated ED presentations [29].

In conclusion, our findings provide evidence of excess repeat acute care in both children and adults with intellectual disability and epilepsy in a universal healthcare setting. They suggest the need for interdisciplinary team management in the care of people with intellectual disability and epilepsy in both hospital and community settings. We recommend further research on strategies to improve management of seizures and comorbidities in people with intellectual disability and epilepsy. Future research should also identify drivers of repeat acute care, particularly preventable factors, and how they differ between people with and without intellectual disability. As a result of continuous deinstitutionalisation, people with epilepsy and intellectual disability increasingly attend mainstream healthcare services. Failure to understand and cater to their needs may inadvertently widen the health disparities experienced by people with intellectual disability.

## Supporting information

**S1 Table. Variables and respective values used to define an unplanned hospitalisation or readmission.**
(DOCX)

**S2 Table. Baseline characteristics of children and adults, including those with incomplete data, with and without intellectual disability (n, %).**
(DOCX)

## Author Contributions

**Conceptualization:** Peiwen Liao, Claire M. Vajdic, Simone Reppermund, Rachael C. Cvejic, Tim R. Watkins, Preeyaporn Srasuebkul, Julian Trollor.

**Data curation:** Preeyaporn Srasuebkul.

**Formal analysis:** Peiwen Liao.

**Funding acquisition:** Claire M. Vajdic, Simone Reppermund, Julian Trollor.

**Methodology:** Peiwen Liao, Claire M. Vajdic, Simone Reppermund, Rachael C. Cvejic, Tim R. Watkins, Preeyaporn Srasuebkul, Julian Trollor.

**Supervision:** Claire M. Vajdic, Simone Reppermund, Rachael C. Cvejic, Julian Trollor.

**Writing – original draft:** Peiwen Liao.

**Writing – review & editing:** Peiwen Liao, Claire M. Vajdic, Simone Reppermund, Rachael C. Cvejic, Tim R. Watkins, Preeyaporn Srasuebkul, Julian Trollor.

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
