## [Decision Letter · Decision Letter 0]

17 May 2022

PONE-D-22-10686Readmission and emergency department presentation after hospitalisation for epilepsy in people with intellectual disability: a data linkage studyPLOS ONE

Dear Dr. Vajdic,

Thank you for submitting your manuscript to PLOS ONE. After careful consideration, we feel that it has merit but does not fully meet PLOS ONE’s publication criteria as it currently stands. Therefore, we invite you to submit a revised version of the manuscript that addresses the points raised during the review process. Please take a close look at the comments from the two reviewers. Reviewer #2 feels that a flowchart might help in presenting procedure and data. Reviewer #1 comments on the justification of conclusions and suggests to clarify terms (e.g. neurodevelopmental). 

We look forward to receiving your revised manuscript.

Kind regards,

Robert Didden

Academic Editor

PLOS ONE

Journal Requirements:

a) Did participants provide their written or verbal informed consent to participate in this study?

[NO authors have competing interests]. 

Reviewers' comments:

Reviewer's Responses to Questions

**Comments to the Author**

1. Is the manuscript technically sound, and do the data support the conclusions?

Reviewer #1: Yes

Reviewer #2: Partly

2. Has the statistical analysis been performed appropriately and rigorously? 

Reviewer #1: I Don't Know

Reviewer #2: I Don't Know

3. Have the authors made all data underlying the findings in their manuscript fully available?

Reviewer #1: Yes

Reviewer #2: No

4. Is the manuscript presented in an intelligible fashion and written in standard English?

Reviewer #1: Yes

Reviewer #2: No

5. Review Comments to the Author

Reviewer #1: The authors should avoid making dogmatic statements that appear to be conclusions from the data but are actually speculation. For example, (page 2, lines 46 & 47) the authors are not justified in stating that “Increased risk of returning to acute care after epilepsy hospitalisation indicates an urgent need to improve epilepsy care for people with intellectual disability.” Although it might be reasonable to assume that improving epilepsy care would decrease the risk of returning to acute care, this is an assumption. A statement such as: “Increased risk of returning to acute care after epilepsy hospitalisation suggests that there is a need to improve epilepsy care for people with intellectual disability”, or similar would, however, be acceptable.

Similarly (page 2, lines 47 et seq.) the authors are not justified in stating “Our findings highlight the importance of both seizures and comorbidity management in reducing recurrent acute care in people with both epilepsy and intellectual disability.” Their findings do, indeed, suggest that management of these conditions might be important in decreasing readmission but they have not shown this. They might have written something along the following lines.

“We recommend research into strategies of improving management of both seizures and comorbidity, with the aim of decreasing readmission.”

Page 7. Line 142. There appears to be a stray bracket before “length of stay”.

Page 8. Line 149. The reader might be wondering what the “other neurodevelopmental conditions” might be. Is there any reason why the authors have not stated what these are or, at least, why they have not provided examples such as “autism and attention deficit hyperactivity disorder”?

Page 12. Lines 196 et seq. The reported psychiatric comorbidities in adults with and without intellectual disability were remarkably similar. In contrast, the psychiatric comorbidities in children with and without intellectual disability were markedly different. The authors have not drawn attention to this striking result or suggested reasons for it. They should do so.

Page 16.

Line 251. The construction “without within” is very clumsy. It could be improved to some degree by at least inserting a comma: “without, within”.

Line 259. Where the authors have written “RR within 365 days” did they mean to write “the RR within 365 days”?

Page 17. Lines 268 et seq. This is another example of the authors drawing unjustified conclusions from their results. They should be less dogmatic. They could have written something along the following lines.

“These findings suggest that there is a need to provide specialised….”

Page 18.

Lines 311 et seq. Again, the authors are speculating and should be less dogmatic. Instead of writing: “This emphasises the need”, the authors could have written: “This suggests that there is a need”. Similarly, instead of writing “should emphasise”, they could have written “might emphasise”.

Page 19.

Lines 319 & 320. Again, the authors appear to be drawing an unjustified conclusion. Instead of writing: “This highlights the need”, they could have written: “This suggests that there is a need”.

Line 399. Where the authors have written “diagnoses” did they mean to write “the diagnosis”. If, which appears unlikely, they did mean to write the plural “diagnoses” then this should be followed by the plural “were”.

Reviewer #2: This is a substantial study, but I found the data presentation so confusing that I am unwilling to comment on the extent to which it addresses the underlying question-do people with intellectual disability have a problem with the response to their condition over and above any intrinsic problems related to the condition itself. What is needed is a flow chart that starts with the total population in the linked data base and comparators, and then flows through to identify the base and study populations involved and the comparators, in such a way that it is easy to track all the relevant populations studied. Wihout that, I am wary of commenting on the conclusions.

On a more general note, the data linkage population has already been used to demonstrate that patients will intellectual disability have a problem related to hospitalisations. Studies that try to understand what happens to patients with intellectual disability, and how that differs from the management in the absence of intellectual disability would be very useful at this point.

6. PLOS authors have the option to publish the peer review history of their article (what does this mean?). If published, this will include your full peer review and any attached files.

Reviewer #1: No

Reviewer #2: **Yes: **Professor David Ben-Tovim

---

## [Author Response · Author response to Decision Letter 0]

25 Jun 2022

From academic editor:

Journal Requirements:

Response: Thank you. We have adjusted the format accordingly. 

a) Did participants provide their written or verbal informed consent to participate in this study?

Response: Thank you. The requirement for informed consent was waived by the presiding ethics committee because the research fulfilled the exemption requirements. Critically, the data shared with researchers was de-identified and additional safeguards were in place to protect the privacy and confidentiality of individuals and institutions. The ethics statement has been revised:

Page 8. Lines 161-163: “……Committee, and access to the data sets was granted by relevant data custodians. As the research met specific safeguards, the requirement for informed consent was waived by the presiding ethics committee.”

[NO authors have competing interests]. 

Response: Thank you. The authors have declared that no competing interests exist.

Response: Thank you. The original data are not publicly accessible due to the conditions of use imposed by the data custodians and ethics committee. We revised the data availability statement as follows: 

Page 8. Lines 165-168: access to the data and analytical files is permitted subject to the approval of the human research ethics committees and data custodians. Researchers interested in applying for data access or collaboration should contact the Department of Developmental Disability Neuropsychiatry (dddn@unsw.edu.au) with their expression of interest.

From reviewer 1:

Reviewer #1: The authors should avoid making dogmatic statements that appear to be conclusions from the data but are actually speculation. For example, (page 2, lines 46 & 47) the authors are not justified in stating that “Increased risk of returning to acute care after epilepsy hospitalisation indicates an urgent need to improve epilepsy care for people with intellectual disability.” Although it might be reasonable to assume that improving epilepsy care would decrease the risk of returning to acute care, this is an assumption. A statement such as: “Increased risk of returning to acute care after epilepsy hospitalisation suggests that there is a need to improve epilepsy care for people with intellectual disability”, or similar would, however, be acceptable.

Similarly (page 2, lines 47 et seq.) the authors are not justified in stating “Our findings highlight the importance of both seizures and comorbidity management in reducing recurrent acute care in people with both epilepsy and intellectual disability.” Their findings do, indeed, suggest that management of these conditions might be important in decreasing readmission but they have not shown this. They might have written something along the following lines.

“We recommend research into strategies of improving management of both seizures and comorbidity, with the aim of decreasing readmission.”

Response: Thank you for this constructive comment. We have revised the sentence accordingly:

Pages 2-3. Lines 46-50: “Increased risk of returning to acute care after epilepsy hospitalisation suggests there is a need to improve epilepsy care for people with intellectual disability. We recommend research into strategies to improve management of both seizures and comorbidity.”

Page 7. Line 142. There appears to be a stray bracket before “length of stay”.

Response: Thank you for bringing this to our attention. We have corrected the mistake and re-organised the brackets. 

Page 8. Lines 146-148: “…… characteristics of the index admission (length of stay, hospital type [public, private], mode of discharge [community, other accommodation or nursing home]), and comorbidity status as covariates.”

Page 8. Line 149. The reader might be wondering what the “other neurodevelopmental conditions” might be. Is there any reason why the authors have not stated what these are or, at least, why they have not provided examples such as “autism and attention deficit hyperactivity disorder”?

Response: Thank you for bringing this to our attention. We have added the examples as suggested. 

Page 8. Lines 153-154: “……are other neurodevelopmental conditions (e.g., autism and attention deficit hyperactivity disorder) [33].”

Page 12. Lines 196 et seq. The reported psychiatric comorbidities in adults with and without intellectual disability were remarkably similar. In contrast, the psychiatric comorbidities in children with and without intellectual disability were markedly different. The authors have not drawn attention to this striking result or suggested reasons for it. They should do so.

Response: Thank you. In the original manuscript, we did provide potential explanations for this finding (Page 8. Line 152-155): “The most common neuropsychiatric disorders in children with intellectual disability are other neurodevelopmental conditions (e.g., autism and attention deficit hyperactivity disorder) [33], which are less common in the general population [34] and, more importantly, may lie on the causal pathway”. 

We have now added the further explanation on Page 12. Lines 209-214: “…….As noted in the statistical analyses section, the difference in prevalence of psychiatric comorbidities in children with and without intellectual disability may primarily be due to the higher risk of neurodevelopmental disorders in children with intellectual disability Neurodevelopmental conditions (e.g. autism and attention deficit hyperactivity disorder) are typically diagnosed early in life, whereas other mental health conditions (e.g. depression and anxiety) are more likely to be diagnosed during young adulthood [35].”

Reference 35: 

Solmi M, Radua J, Olivola M, Croce E, Soardo L, Salazar de Pablo G, et al. Age at onset of mental disorders worldwide: large-scale meta-analysis of 192 epidemiological studies. Mol Psychiatry. 2022;27(1):281-95. doi: 10.1038/s41380-021-01161-7.

Page 16.

Line 251. The construction “without within” is very clumsy. It could be improved to some degree by at least inserting a comma: “without, within”.

Line 259. Where the authors have written “RR within 365 days” did they mean to write “the RR within 365 days”?

Response: Thank you. We have revised the text accordingly:

Page 16. Lines 267-269: “In both observation periods (30 and 365 days), a greater proportion of people with intellectual disability presented to ED compared to those without (Table 2).”

Page 16. Line 276: “……., all RRs remained significant for adults and only the RR within 365 days remained significant for children.”

We also revised another sentence with a similar structural concern:

Page 16. Lines 261-263: “The age-adjusted model also showed an increased risk of psychiatric readmission within each follow-up period in adults with intellectual disability compared to those without (Table 2), ……”

Page 17. Lines 268 et seq. This is another example of the authors drawing unjustified conclusions from their results. They should be less dogmatic. They could have written something along the following lines.

“These findings suggest that there is a need to provide specialised….”

Response: Thank you for pointing this out. We have revised the sentence:

Page 17. Lines 285-287: “These findings suggest there is a need to provide specialised and interdisciplinary support for inpatient and post-acute care support around epilepsy management for people with intellectual disability.”

Page 18.

Lines 311 et seq. Again, the authors are speculating and should be less dogmatic. Instead of writing: “This emphasises the need”, the authors could have written: “This suggests that there is a need”. Similarly, instead of writing “should emphasise”, they could have written “might emphasise”.

Response: We agree and have revised the sentence:

Page 18. Lines 328-329: “This suggests that there is a need for improved clinical epilepsy care pathways in this high-risk patient group, which might emphasise continuity of care in the community……”.

We also revised another sentence with a similar concern:

Page 19. Line 331: “Disability sector staff could also be better equipped to support……”

Page 19.

Lines 319 & 320. Again, the authors appear to be drawing an unjustified conclusion. Instead of writing: “This highlights the need”, they could have written: “This suggests that there is a need”.

Line 399. Where the authors have written “diagnoses” did they mean to write “the diagnosis”. If, which appears unlikely, they did mean to write the plural “diagnoses” then this should be followed by the plural “were”.

Response: Thank you for your comments. We agree and have revised the text accordingly:

Page 19. Lines 336-337: “This suggests a need for better influenza and aspiration prevention in their epilepsy care.” 

Page 20. Line 357: “Furthermore, as the diagnosis of epilepsy not otherwise specified was common, ……”

We have also revised the conclusion paragraph:

Page 20. Lines 364-370: “They suggest the need for interdisciplinary team management in the care of people with intellectual disability and epilepsy in both hospital and community settings. We recommend further research on strategies to improve management of seizures and comorbidities in people with intellectual disability and epilepsy. Future research should also identify drivers of repeat acute care, particularly preventable factors, and how they differ between people with and without intellectual disability.”

From reviewer 2:

Reviewer #2: This is a substantial study, but I found the data presentation so confusing that I am unwilling to comment on the extent to which it addresses the underlying question-do people with intellectual disability have a problem with the response to their condition over and above any intrinsic problems related to the condition itself. What is needed is a flow chart that starts with the total population in the linked data base and comparators, and then flows through to identify the base and study populations involved and the comparators, in such a way that it is easy to track all the relevant populations studied. Without that, I am wary of commenting on the conclusions.

On a more general note, the data linkage population has already been used to demonstrate that patients will intellectual disability have a problem related to hospitalisations. Studies that try to understand what happens to patients with intellectual disability, and how that differs from the management in the absence of intellectual disability would be very useful at this point.

Response: We would like to thank the reviewer for this input. In response, we have revised the flowchart (Fig 1) to provide information about people with ID and the comparison group. We have also revised the text to further explain the flowchart:

Pages 5-6. Lines 92-100: “Fig 1 describes how the cohort was derived. From all people who were discharged from hospital during the study period, we included people discharged from an unplanned acute care admission between 1 July 2005 and 30 June 2014 where the principal diagnosis recorded was epilepsy (G40) or status epilepticus (G41). Table S1 documents how we defined ‘unplanned’ and ‘acute’. The first admission was defined as the index admission. After excluding individuals with implausible death or age records, we excluded individuals if on discharge from the index admission they i) died, were transferred to hospice care, left hospital against advice, or had no discharge mode recorded [21]; ii) were younger than five or older than 64 years; or iii) had missing covariate data.”

Page 9. Lines 171-178: “Out of 21862 patients with an admission for epilepsy during the study period, 20756 patients were formally discharged from the hospital, among which, we included all patients aged 5-64 years without missing records for any covariates (2574 with intellectual disability; Fig 1). This final cohort consisted of 2104 children and 11433 adults.”

We defined our cohort and the outcomes in alignment with the research question – does having intellectual disability increase the risk of readmission after an unplanned hospitalisation for epilepsy? The inclusion and exclusion criteria implemented are justified by our aim or previous practice. To answer this question more robustly, we have performed separate analyses for children and adults; we have further accounted for the potential confounding effect of age for the adult group. We have also controlled for a variety of potential confounders in the model and acknowledged the potential residual confounding bias in the discussion.

For the second point, we have addressed this point in our revised recommendations for future research (Page 20. Lines 366-369):

“We recommend further research on strategies to improve management of seizures and comorbidities in people with intellectual disability and epilepsy. Future research should also identify drivers of repeat acute care, particularly preventable factors, and how they differ between people with and without intellectual disability.”

---

## [Decision Letter · Decision Letter 1]

20 Jul 2022

Readmission and emergency department presentation after hospitalisation for epilepsy in people with intellectual disability: a data linkage study

PONE-D-22-10686R1

Dear Dr. Vajdic,

We’re pleased to inform you that your manuscript has been judged scientifically suitable for publication and will be formally accepted for publication once it meets all outstanding technical requirements.

Kind regards,

Robert Didden

Academic Editor

PLOS ONE

Additional Editor Comments (optional):

Reviewers' comments:

Reviewer's Responses to Questions

**Comments to the Author**

1. If the authors have adequately addressed your comments raised in a previous round of review and you feel that this manuscript is now acceptable for publication, you may indicate that here to bypass the “Comments to the Author” section, enter your conflict of interest statement in the “Confidential to Editor” section, and submit your "Accept" recommendation.

Reviewer #2: All comments have been addressed

2. Is the manuscript technically sound, and do the data support the conclusions?

Reviewer #2: Yes

3. Has the statistical analysis been performed appropriately and rigorously? 

Reviewer #2: Yes

4. Have the authors made all data underlying the findings in their manuscript fully available?

Reviewer #2: Yes

5. Is the manuscript presented in an intelligible fashion and written in standard English?

Reviewer #2: Yes

6. Review Comments to the Author

Reviewer #2: This article is now ready for publication. I recommend acceptance. I have no further comments to make. The revisions have met this reviewers requirements

7. PLOS authors have the option to publish the peer review history of their article (what does this mean?). If published, this will include your full peer review and any attached files.

Reviewer #2: No

---

## [Editor Report · Acceptance letter]

22 Jul 2022

PONE-D-22-10686R1 

Readmission and emergency department presentation after hospitalisation for epilepsy in people with intellectual disability: a data linkage study 

Dear Dr. Vajdic:

I'm pleased to inform you that your manuscript has been deemed suitable for publication in PLOS ONE. Congratulations! Your manuscript is now with our production department. 

Kind regards, 

on behalf of

Professor Robert Didden 

Academic Editor

PLOS ONE